# Emotion Regulation Difficulties, Family Functioning, and Well-Being Involved in Non-Suicidal Self-Injury and Suicidal Risk in Adolescents and Young People with Borderline Personality Traits

**DOI:** 10.3390/children10061057

**Published:** 2023-06-13

**Authors:** Rosario J. Marrero, Macarena Bello, Daida Morales-Marrero, Ascensión Fumero

**Affiliations:** 1Departamento de Psicología Clínica, Psicobiología y Metodología, Facultad de Psicología y Logopedia, Universidad de La Laguna, 38200 La Laguna, Spain; daidamorales98@gmail.com (D.M.-M.); afumero@ull.edu.es (A.F.); 2Instituto Universitario de Neurociencia (IUNE), Universidad de La Laguna, 38200 La Laguna, Spain; 3Instituto Andrés Bello, 38007 Santa Cruz de Tenerife, Spain; macarena_bello_martin@hotmail.com

**Keywords:** borderline personality disorder traits, non-suicidal self-injury, suicide ideation, suicide planning, suicide attempts, emotion regulation, well-being, family functioning

## Abstract

Individuals with a borderline personality disorder (BPD) or BPD traits usually have a lifetime history of harmful behaviors. Emotion regulation difficulties are a risk factor for suicide, whereas adequate family functioning and well-being play an important protective role. This study aims to determine the role of emotion regulation difficulties, well-being, and family functioning in the suicide risk and non-suicidal self-injury (NSSI) in adolescents and young people with BPD traits. From a sample of 285 young people, 103 (36.1%) had BPD traits (mean age = 16.82, SD = 2.71), and 68.93% were females. The results showed significant differences in personal and family variables according to the type of harmful behavior. Suicide attempts (SA) were mainly predicted by difficulties in impulse control, whereas NSSI was predicted by low family satisfaction. Programs designed to prevent SA and NSSI should consider individual differences, as well as the type of harmful behaviors exhibited.

## 1. Introduction

Adolescence is characterized by intense biological, psychological, and social changes. This developmental stage can generate confusion, weakness, vulnerability, or even serious mental health problems [1]. Evidence suggests a continuum from personality traits dimensions to abnormal personality or disorder, as well as the potential effect on mental health problems throughout life [2]. Although the delimitation of different personality disorders is complex, due to their high comorbidity [3], much clinical research on adolescence has focused on borderline personality disorder (BPD) because of its association with other severe functional impairment and higher risk suicide [4]. In the adolescent population and emerging adults (young people), epidemiological studies suggest a prevalence of around 1–3% for BPD [5]. However, the prevalence of BPD ranges from 11–22% in adolescents and young people at outpatient clinical consultation to 33–49% in inpatients [5]. The prevalence rate of NSSI in adolescents and young people with BPD (15–25-year-olds) is between 76% and 95% [6,7], while the prevalence of suicide attempts in adolescents with BPD is around 78% [8] and 66% in young adults [6].

BPD is considered a severe personality disorder that affects personality functioning (identity problems or instability in self-direction), interpersonal functioning (problems with intimacy or empathy), and at least four other pathological personality traits, including negative affectivity (emotional liability, anxiousness, separation insecurity, and depressivity), disinhibition (risk-taking or impulsivity), and antagonism (hostility) [9]. Although reviews have found that the characteristics of BPD are comparable between adolescents and adults, the diagnosis of BPD in adolescence and younger individuals remains controversial [10]. Findings highlight that subthreshold BPD in adolescents displaying risk-taking and self-harming behaviors is even associated with impairments comparable to full-syndrome BPD [4] and support a lower diagnostic cut-off for adolescent BPD to identify those at risk at an early stage. Consequently, BPD traits could be used more effectively for research purposes.

Linehan’s biosocial model [11,12] proposes that individuals with BPD symptoms show biological vulnerability and high negative emotional intensity that interact with environmental risk factors, such as adverse childhood experiences or attachment difficulties. Both emotion dysregulation and family relationships have been considered as a main mechanism in BPD. Some of the most prominent BPD symptoms, including affective instability, separation anxiety, or impulsivity, have been shown to be associated with non-suicidal self-injury (NSSI) in a large sample of Belgian adolescents [13]. Specifically, the difficulties in affective and behavioral emotion regulation represent a diagnostic core feature of adolescent BPD [8,14]. The experiences of chronic emotion regulation difficulties or invalidation of emotions by adolescents with BPD might be triggering suicide ideation (SI), suicide attempts (SA) [8,12], and NSSI [15]. In this sense, the affect regulation model of NSSI indicates that it helps to regulate emotions by reducing aversive internal states or emotions such as sadness, anger, or emptiness and to avoid interpersonal demands such as social rejection, the ultimate goal being to gain attention or increase social support [16]. On the other hand, the motivational–volitional model of suicidal behavior proposes that negative self-evaluations cause the person to feel trapped and to contemplate suicide as a possible strategy to regulate their negative emotions [17]. Mediation studies have found a relationship between emotion regulation deficits, NSSI, and BPD features in Italian adolescents, although emotional dysregulation is not the only factor explaining the association between NSSI and BPD [14].

Family factors and interpersonal childhood adversities have also been shown to contribute to developing and maintaining BPD [18,19]. Specifically, lack of involvement, family conflict, or lack of regard have been linked to BPD traits in the offspring of adoptive families with familial risk factors, such as previous parental psychopathology, while genetic transmission of BPD traits has been found in biological parents [20]. Other family factors involved in BPD symptoms are chaotic parenting, which can lead to emotional dysregulation and impulsivity; controlling parenting practices, which can lead to a lack of confidence in relationships; or rejecting parenting practices, which has been associated with identity disturbances and unstable relationships in Canadian adolescents [21]. In a qualitative study with young women aged between 21 and 37 years recently diagnosed with BPD, the issues that most concerned them were related to feelings of insecurity, self-distrust, and dependence on others to feel safe. These experiences were managed through addictive behavior, NSSI, and SA [22]. Poor quality family interaction and communication and a lack of emotional expression in the family context are associated with an increased risk of suicidal ideation, whereas participation in family leisure activities decreases that risk in 14- to 18-year-old Spanish and Latin adolescents with BPD traits [23]. Additionally, in a community sample of Chilean adolescents, a higher suicide risk is associated with perception of the family as dysfunctional, exposure to physical or psychological aggression, lower self-esteem, poor social support, mistreatment by a partner, and lower health-related quality of life [24]. Despite the potential role of well-being in preventing mental health problems, mainly through emotional regulation [25], research on well-being in BPD, NSSI, and suicidality is scant. Results are mostly along the lines of a detriment in well-being in undergraduate students with BPD traits [26] and lower well-being among community adolescents at risk of suicide [27].

The clinical profile of inpatient adolescents with BPD is unique, suggesting more serious suicidal behavior compared to other adolescents without BPD who attempt suicide [28]. The peak age at which NSSI appears is 14–15 or 20–24 years for the community population and the clinical sample [29]. Both NSSI and SA decrease throughout adolescence [30,31]. Although BPD may lead to a higher risk of both SA and NSSI, the relationship between SA and NSSI is complex. A strong correlation between SA and NSSI behaviors has been found in community adolescents and young adults with BPD traits [32,33]. Therefore, emotion regulation difficulties, negative family functioning, and low well-being have been shown to affect the risk of NSSI and suicide behavior in individuals with BPD [34]. However, few studies have examined the combined impact of emotion regulation difficulties and family functioning in NSSI and the different suicidal behaviors (ideation, planning, and attempt) in a community sample of young people with and without BPD traits. 

The objectives of this study were two-fold: (1) to analyze the differences in self-harm behaviors (NSSI, SI, planning, and attempts), personal variables (emotion regulation difficulties and well-being), and family variables (family cohesion, adaptability, and satisfaction) between adolescents and young people with and without BPD traits; and (2) to determine the weight of personal, family/interpersonal, and well-being variables on the risk of NSSI and the different suicide indicators in adolescents and young people characterized by borderline personality traits, by controlling the effect of sociodemographic variables (gender and age). 

**Hypothesis** **1.**
*Adolescents and young people with BPD traits are expected to show higher NSSI and suicidal risk, higher emotion regulation difficulties, and lower family adjustment and lower well-being than those without BPD traits.*


**Hypothesis** **2.**
*On the other hand, it is expected that a differential profile will be found at both the emotional and family level to explain the different NSSI and suicidal behaviors of adolescents and young people with BPD traits.*


## 2. Materials and Methods

### 2.1. Participants

Survey respondents were 285 young people (12–25 years) residing in the Canary Islands. The mean age of the sample was 16.82 (SD = 2.71), with 66.3% (n = 189) identifying as female, 32.3% (n = 92) as male, and 1.4% (n = 4) not responding. Participants reported age as follows: 12–14 years (25.26%, n = 72), 15–17 (28.07%, n = 80), and 18–25 (39.65%, n = 113). 

From this sample, 103 participants (36.14%) with BPD traits were selected: 68.9% girls (n = 71), 27.18% (n = 28) boys, and 3.88% (n = 4) who did not respond. The participants who did not respond to the gender question belonged to the BPD traits group. The age groups were as follows: 12–14 years (33.98%, n = 35), 15–17 (25.24%, n = 26), and 18–25 (40.78%, n = 42). The age group with the highest mean in borderline trait was 12–14 years ($\overset{-}{x}$ = 2.33; SD = 1.43).

### 2.2. Instruments

A demographic form enabled participants to provide demographic information on age, gender, and level of education.

The Paykel Suicide Scale [35] is a 5-item self-report tool to assess suicide ideation (SI, 2 items), suicide planning (SP, 2 items), and suicide attempt (SA, 1 item), with a dichotomous (Yes or No) response format. Higher scores indicate greater suicidal risk. A global score of suicidal risk can be obtained by adding the affirmative answers to the five items. One open question about the reasons for SA was included. The tool has shown adequate psychometric criteria (Cronbach’s alpha of 0.93) in Spanish young people [36]. In this study, Cronbach’s alpha was 0.93.

A 4-item questionnaire was used for NSSI [37]. Two items assessed ideation and NSSI attempts in a dichotomous (Yes or No) response format, and another two open questions explored the type of self-injury behavior and the reasons for it. In this study, only items about NSSI attempts and the reasons for them were examined.

The International Personality Disorders Examination [38,39] assesses the presence of pathological personality characteristics according to the International Statistical Classification of Diseases and Related Health Problems-10 [40]. The self-administered questionnaire consists of 59 items, with a true/false response format. The BPD subscale includes five items. A score of three or more on each scale indicates that participants have failed the screen for that disorder. For this study, a diagnostic subthreshold of BPD or BPD traits defined as the occurrence of three positive items was adopted. This cut-off point has been described as the best combination of sensitivity and specificity [41]. In this research, the self-administered screening questionnaire was used to assess the presence of BPD traits. Cronbach’s alpha was 0.50 for this study.

The Difficulties in Emotion Regulation Scale [42] is a 36-item self-report questionnaire that clinically measures relevant aspects of emotional dysregulation. It is composed of six subscales: Lack of emotional awareness (6 items), Lack of emotional clarity (5 items), Impulse control difficulties (6 items), Difficulty engaging in goal-directed behavior (5 items), Non-acceptance of negative emotional responses (6 items), and Limited access to emotion regulation strategies (8 items). A 5-point Likert scale ranges from 1 (almost never) to 5 (almost always), with higher scores indicating greater emotion regulation difficulties. The Spanish adolescent version obtained the following Cronbach’s alpha: Lack of emotional awareness (0.62), Lack of emotional clarity (0.71), Impulse control difficulties (0.81), Difficulty engaging in goal-directed behavior (0.80), Non-acceptance of negative emotional responses (0.84), and Limited access to emotion regulation strategies (0.77), respectively [43]. The adult version also has adequate psychometric criteria ranging from 0.68 for lack of emotional awareness to 0.90 for limited access to emotion regulation strategies [44]. In the current study, the adolescent version was used for all participants, and the factors showed the following internal consistency: Lack of emotional awareness (0.73), Lack of emotional clarity (0.86), Impulse control difficulties (0.85), Difficulty engaging in goal-directed behavior (0.84), Non-acceptance of negative emotional responses (0.93), and Limited access to emotion regulation strategies (0.85), respectively.

The Personal Well-being Index school version [45] assesses satisfaction in different aspects or life situations. Because the child and adult versions of the well-being index are equivalent [46], the same version was applied to all participants. The index consists of seven items that assess well-being in seven vital areas (standard of living, personal health, achievement in life, personal relationships, personal safety, feeling part of the community, and future security); an 11-point Likert-type response scale, ranging from totally dissatisfied (0) to totally satisfied (10), was used. The items are averaged to obtain a total score, which represents personal satisfaction. The Cronbach’s alpha of the Spanish version was 0.82 [47], and in this study, it was 0.83.

The Family Adaptability and Cohesion Evaluation Scale [48] assesses the cohesion and adaptability of family functioning. Cohesion is the emotional bond between the members of a system, and adaptability is the ability of this system to change. The Cronbach’s alpha in a Spanish sample was 0.89 for cohesion and 0.87 for adaptability. In this study, it was 0.92 and 0.94 for cohesion and adaptability, respectively.

The family satisfaction scale (self-scale questionnaire) was designed for this research and includes only one item with a 5-point Likert-type response scale, ranging from totally dissatisfied (1) to totally satisfied (5). The specific question asked was as follows: How would you rate your overall level of satisfaction with your family?

### 2.3. Procedure

To recruit adolescent participants, five state schools in Tenerife were invited to take part: three from the metropolitan area, one from the south of the island, and one from the north. Only two schools from the metropolitan area agreed to participate. To make the sample more representative, the link to the questionnaires was sent to parents of teenage children in the researchers’ circle to create a snowballing effect on adolescents from other schools. Firstly, the Pedagogical Coordination Commission and school governance boards of each school were informed about the research and were shown the scales of the student evaluation protocol. Once participation was requested and accepted by both parents and students through informed consent, meetings were organized for each group to explain the research and instruct them how to complete the questionnaire, through a voluntary and confidential online protocol. Students agreeing to participate in the research completed the evaluation protocol online in the classroom supervised by a teacher who answered any queries. The entire evaluation protocol took approximately one hour to complete. Participants over 18 years of age were psychology undergraduates who received an extra grade for participating and circulating the evaluation protocol to other students at the same university. The instruments were completed online at home. Anonymity and confidentiality were guaranteed at all times. Data were collected from adolescents and university students from November 2020 to March 2021. All procedures received approval from the university’s Institutional Review Board (CEIBA 2020-0412) (6 July 2020), and informed consent was given. 

### 2.4. Data Analysis

The statistical analyses were conducted using IBM SPSS Statistics, version 25. First, chi-square differences between the sample with and without BPD traits were analyzed to detect sample homogeneity for age and gender. Second, MANCOVA was used to determine whether there were statistically significant differences between the groups with and without BPD traits in all variables of interest, after controlling for gender and age. Third, bivariate correlations among the variables of suicide risk and NSSI in the BPD traits group were used to check co-occurrence between variables. Fourth, a frequency analysis of reasons for NSSI and SA was conducted. Fifth, logistic regression was applied to predict suicide risk (SI, SP, and SA) and NSSI in young people with borderline traits, using the personal and family/interpersonal variables of the study as predictors. G*Power [49] was used for post hoc analyses of the statistical power of the results from the available sample. 

## 3. Results

### 3.1. Preliminary Analysis

The chi-square analysis showed statistically significant differences in age (X^2^(2) = 6.51, *p* = 0.039) and gender (X^2^(2) = 8.53, *p* = 0.014) between adolescents and young adults with and without BPD traits. The adolescents and young adults with BPD traits were mainly older girls.

### 3.2. Mean Differences between the Group with and without BPD

When controlling for gender and age, the MANCOVA results showed differences in all the variables of interest, except in Lack of emotional awareness, for the group with BPD traits compared to the group without BPD traits. Effect size was medium for family functioning, satisfaction, and SA and large for suicide risk and NSSI, emotion regulation difficulties, and well-being (see Table 1).

### 3.3. BPD Traits and NSSI and Suicide Risk

Pearson correlations among the variables of NSSI and suicide risk in the BPD traits group were significant (*p* < 0.001); r = 0.36 between NSSI and SI; r = 0.47 between NSSI and SP; and r = 0.53 between NSSI and SA.

Frequency analysis indicated that, of the total of adolescents and young people with BPD traits, 35% reported the reasons for NSSI. Of these, the most common reasons were to “punish myself for feeling unwell” (27.8%), negative mood such as depression (25%), family difficulties (16.7%), “I wanted to die” (8.3%), and feeling lonely (5.6%). The remaining reasons were related to unspecified personal experiences or other adverse situations. On the other hand, of the total number of adolescents and young people with BPD, 20.4% reported the reasons for SA. Of these, the most common were anxiety (23.8%), feeling that they do not fit in anywhere (23.8%), family difficulties (19%), and not feeling pain (14.3%). The remaining reasons were related to feeling worthless.

A logistic regression analysis was applied to identify which variables classified adolescents and young people with BPD traits into suicidal ideation, planning and attempts, and self-injury groups. Gender and age were controlled. Gender consisted of three categories: 0 male, 1 female, and 2 prefer not to answer. Two dummy variables (k-1) were therefore created in which the reference value was male, value 1 corresponded to female, and value 2 to those who prefer not to answer. Table 2 and Table 3 show the results of the logistic regression analyses.

The results indicated that low family satisfaction and lack of emotional clarity were associated with SI. These variables explained 34% of the variance of SI. The percentage of correctly classified cases was 95.1% for adolescents and young people with SI and 42.9% for those without SI. Post hoc analyses indicated that the statistical power (1-β = 0.13) was not high enough.

Limited access to emotion regulation strategies and low well-being were the significant predictor variables that explained 41% of the variance of the SP. The percentage of correctly classified cases was 91.4% for adolescents and young people planning suicide and 72% for those who were not. The post hoc statistical power was low (1-β = 0.14).

Difficulties in impulse control were the only significant predictor for the SA. This variable explained 29% of the variance of SA, although the percentage of correctly classified cases was 51.9% for adolescents and young people with SA and 90.8% for those without SA. Post hoc analyses of statistical power (1-β = 0.11) indicated a limited role for the results obtained.

Furthermore, 25% of the variance of NSSI was explained by lower family satisfaction. The percentage of correctly classified cases was 60.9% for adolescents and young people with NSSI and 73.7% for those with none. The post hoc statistical power (1-β = 0.73) also failed to reach an acceptable level.

## 4. Discussion

The main goal of this study was to examine the contribution of emotion regulation difficulties, family functioning, and well-being to NSSI and suicidal behaviors in adolescents and young people with BPD traits. The results provided empirical support for Hypothesis 1, which was the notion that emotion regulation difficulties, family functioning, and well-being are relevant factors associated with BPD traits versus no BPD traits, with an increased risk of NSSI and suicidal behavior also appearing. In parallel, a profile difference was found through which individuals with BPD traits may experience one or other type of suicidal or NSSI behaviors, thereby confirming Hypothesis 2. NSSI was mainly predicted by low family satisfaction; suicidal ideation was predicted by lack of emotional clarity and low family satisfaction; suicidal planning was predicted by limited access to emotion regulation strategies and low well-being; and suicidal intent was predicted by impulse control difficulties.

Adolescents and young people with BPD traits showed higher suicide risk and NSSI, as well as greater difficulty regulating emotions, lower family cohesion, adaptability and satisfaction, and lower well-being than adolescents and young people without BPD. There is considerable evidence that emotion regulation skill deficits may differentiate individuals with BPD from other samples and are associated with higher BPD symptom severity [50]. Accordingly, findings indicate that emotion regulation problems are related to higher levels of BPD features and symptoms in undergraduate, healthy control, and clinical samples [11]. That emotion regulation difficulties may functionally link all of the BPD symptoms has been suggested [51]. Emotional vulnerability (low emotional threshold and intense emotional response) and an invalidating environment (rejection of or inattention to a child’s emotional experiences) could underlie the development and maintenance of BPD [52].

The prevalence of SI, SP, SA, and NSSI in the sample of adolescents and young people with BPD traits was 79.6%, 67.9%, 26.2%, and 44.7%, respectively. SA in adolescents with BPD traits were similar to those recorded by previous research (77.6%), while the prevalence rate of NSSI was higher in the literature (64.2%) [53]. These findings suggest that NSSI and suicide behaviors are associated with high clinical severity. Additionally, in the current study, the co-occurrence of NSSI and SA was 23.30%; in fact, only 8% of young people reported “wanting to die” among the reasons for NSSI. In previous research with clinical samples of adolescents, between 14% and 70% report both NSSI and SA [32]. Although it has been suggested that NSSI may be a precursor of suicide [54], the co-occurrence rates found in this study do not corroborate this issue. Nevertheless, NSSI and suicide behaviors could be potential markers in the early detection of BPD symptomatology during adolescence [12].

The present study found that in adolescents and young people with BPD traits, emotion regulation difficulties differentially predicted the various types of harmful behaviors. Although it did not reach the significance threshold of 0.05, paradoxically, low difficulty in accepting negative emotions predicted an increased risk of NSSI in adolescents with BDP traits. These findings are inconsistent with those arising from examining the relations between emotion regulation difficulties and NSSI symptoms, which found that emotion regulation difficulties predicted future NSSI, and that this relation is bidirectional [55]. Adolescents and young people with BPD traits in our sample were able to identify and internalize negative emotions, and this emotional distress could lead to self-harm. Adolescents and young people with NSSI reported that negative emotions involved in personal problems, depression, or the need to punish themselves were among the most important reasons for such behavior. These reasons could trigger the discomfort they feel and precede NSSI. NSSI may represent a strategy to regulate the negative emotions experienced. Nock and Prinstein’s model [16] suggests that NSSI has automatic negative intrapersonal functions when NSSI reduces aversive internal states, followed by automatic positive ones when NSSI produces positive internal states.

On the other hand, the non-acceptance of negative emotional responses showed an inverse pattern for SI, though it was only marginally significant. This strategy could be key to distinguishing NSSI behavior from suicidal behavior. These results can be explained by the Integrated Motivational-Volitional Model of Suicidal Behavior (IMV) [17]. Adolescents and young people with SI may not be able to identify what is happening to them, hence their difficulty in accepting negative emotions. In fact, among the reasons they reported for thinking about or attempting suicide were wanting to stop feeling emotional pain and feeling they did not fit in anywhere. Likewise, these feelings could be associated with various stressors, among which could be low satisfaction with family relationships—another reason reported by young people—and high levels of anxiety. These different factors could generate a feeling of entrapment and lead to SI. However, these results should be taken with caution, as the strategy of non-acceptance of negative emotional responses was a weak predictor of both NSSI and SA.

Furthermore, it is interesting to note that the emotion regulation difficulties that predicted SP and SA did not coincide with those involved in SI, lending support to the IMV model of O’Connor et al. [17] and suggesting differences between SI, SP, and SA. In our study, young people with BPD traits who planned suicide mainly showed a lack of emotional clarity, limited access to regulation strategies (not knowing how to regulate), and low well-being. In a community youth sample, the lack of emotional clarity and restricted emotion regulation strategies predicted SA above other factors, such as depression, traditionally involved in this type of behavior [56,57]. Similarly, a lack of access to strategies has been associated with SI in adolescents [58]. As for the adolescents and young people who attempted suicide in our study, the only strategy involved was their inability to control impulses. These findings suggest the idea of a continuum, as proposed by the Three-Step Theory of Suicide [59]. In the first step, suicide is described as a combination of intense pain and hopelessness (reasons given by our participants), which may lead to a desire for suicide or to SI. In a second step, this desire for suicide may increase if the person feels unconnected to others or finds no purpose in life (low family satisfaction). In the third step, a person can move from desire for suicide to an attempt when the capacity for SA has developed, either through tolerance of pain or loss of fear of death.

Although measures of family functioning, cohesion, and adaptability were not significant in predicting suicidal and NSSI behaviors, adolescents and young people with BPD traits reported low family satisfaction, which contributed to predicting both SI and NSSI. Previous research has shown that individuals with BPD report problems with social connectedness [60]. Negative interactions between adolescents and their mothers are linked to BPD symptoms, while supportive interactions have no impact on BPD [61]. Furthermore, an insecure attachment style has been associated with SA through emotion regulation skills in adolescents with BPD [62]. Despite controversy about this issue, it appears that connection and support can facilitate recovery for people with BPD [63].

The current study suggests that, in the face of personal and family/interpersonal problems, if adolescents and young people with BPD traits have no adaptive emotion regulation strategies (emotional clarity, access to emotion regulation strategies, or impulsive behavior control), SI or NSSI behaviors can escalate into SA. However, acceptance of negative emotions when accompanied by NSSI will have an automatic negative reinforcement function, allowing for a reduction in aversive internal states. The differential role these emotion regulation difficulties may have for NSSI and the different stages from SI to SA make it particularly interesting to analyze in depth the reasons that drive adolescents and young people with BPD traits to engage in this type of self-harming behavior. Specifically, the directionality of this relationship should be explored to identify whether emotion regulation difficulties are intrapersonal characteristics that contribute to perceiving others as threatening or, conversely, whether others (e.g., parents or peers) showing behaviors of criticism, withdrawal, or lack of support intensify this lack of connection, thereby increasing interpersonal/family problems and emotion regulation difficulties. 

In this regard, interventions for young people that integrate intrapsychic and interpersonal/family factors have demonstrated their effectiveness in reducing BPD symptoms. Specifically, cognitive analytic therapy, emotion regulation training, systems training for emotional predictability and problem solving, mentalization-based therapy, cognitive behavioral therapy, and dialectical behavior therapy have been shown to be beneficial, though with mixed results [64,65]. All of the interventions share a number of characteristics that have been associated with effective outcomes. The interventions aimed at modifying clinically significant features of BPD have been applied according to developmental stage, and the family or caregivers are actively involved in the treatment [66]. The use of pharmacotherapy with adolescents is not recommended [8]. Both mood stabilizing and second-generation antipsychotics should only be used for treating specific symptoms, such as suicide crisis, NSSI, affective instability, or impulsivity [67]. Medication must complement active collaboration in psychotherapy, which has been shown to be the necessary condition to develop new strategies [68]. Moreover, with adolescents, the disorder must be approached preventatively by identifying family risk factors, such as personality disorder or substance abuse, and addressing the impact on the family of their children’s emotional health problems [67]. Finally, it is important to address not only the relationship with parents but also interaction with peers, siblings, and/or other adults to minimize stressors and to work on problem-solving skills [8].

This study has some limitations. First, the cross-sectional design restricts conclusions about the role of emotion regulation difficulties and familial satisfaction in the development of NSSI and suicide behaviors in adolescents and young individuals with BPD traits. Second, the sample was taken from educational centers on the island of Tenerife, but a representative sample is needed to generalize the results. Post hoc analyses suggest that the results obtained have insufficient statistical power, and a larger sample may have yielded different results. Third, the use of self-reports to record both BPD traits and NSSI and suicide behaviors may bias the information collected. Fourth, adolescents and young people with BPD traits were identified according to a categorical rather than a dimensional model. There is some controversy about how to identify individuals with personality disorders and whether these types of disorders can be identified at an early age. However, some studies indicate that the prevalence of personality disorders in adolescence persists into adulthood, providing validity for the diagnosis [69]. Finally, that BPD shows comorbidity with both internalizing (e.g., mood disorders) and externalizing problems (e.g., attention-deficit and hyperactivity disorder, ADHD) throughout early development interactions occurring between lower-level symptoms should be considered [70,71]. Externalizing disorders during childhood are mainly linked to BPD development in early adolescence, while adolescent internalizing disorders are associated with BPD in adulthood [8]. In addition, other self-harming behaviors, such as substance use, have been shown to be associated with suicidal behavior, NSSI, and BPD traits in adolescents [31]. ADHD symptoms or other comorbid mental disorders have not been explored in this sample; the results should therefore be taken with caution.

Future research should incorporate timeline procedures. In addition, structured clinical interviews to assess BPD symptoms based on dimensional models of BPD could be more useful in adolescents to account for the variability that can appear at this stage of development. Additionally, it would be interesting to explore how emotion regulation deficits relate differentially to SI and SA. Another promising line of future research could be to focus on early intervention strategies aimed at decreasing emotion regulation difficulties and strengthening individuals’ connections to others, thereby decreasing suicide risk among adolescents and young people. A recent study has found that the search for meaning in life is an important protective factor against self-injurious behaviors [72].

## 5. Conclusions

In sum, NSSI and suicidal behaviors are important risk behaviors in adolescents and young people with BPD traits. These signs could be monitored for the early detection of BPD symptoms, which can also be an indicator of NSSI and suicide behaviors. Moreover, emotion regulation difficulties and family relationships have been linked to these risk behaviors. In this sense, risk could be reduced if affective or family instability were identified in adolescence. On the other hand, there may be a continuum from ideation through planning to attempt, although the difficulties in emotion regulation that predominate at each stage have not been shown to be the same. It is striking that the difficulties of impulse control were the most important factor for suicide attempt. However, other emotion regulation strategies, family satisfaction, and subjective well-being in the previous stages should be considered as therapeutic components to prevent suicide.

## Figures and Tables

**Table 1 children-10-01057-t001:** Mean differences between the groups with and without BPD traits.

	Participants with BPD Traits		Participants without BPD Traits					
	M	S.D.	M	S.D.	F	*p*	η²	1-β
Non-suicidal self-injury	0.45	0.50	0.10	0.31	48.84	<0.001	0.15	1
Suicidal ideation	1.44	0.81	0.49	0.80	94.45	<0.001	0.25	1
Suicidal planning	1.10	0.86	0.31	0.62	76.88	<0.001	0.22	1
Suicide attempts	0.26	0.44	0.05	0.23	23.27	<0.001	0.08	0.998
Awareness	18.51	4.81	17.57	4.76	1.86	0.17	0.01	0.275
Impulse	17.73	5.95	12.36	4.89	65.72	<0.001	0.19	1
Non-acceptance	22.50	8.72	14.23	6.70	83.48	<0.001	0.23	1
Goals	18.50	4.55	14.35	5.01	50.22	<0.001	0.15	1
Clarity	16.77	4.68	11.79	4.55	75.82	<0.001	0.21	1
Strategies	21.91	6.17	15.30	5.71	86.79	<0.001	0.24	1
Family cohesion	29.96	9.45	36.68	8.98	35.63	<0.001	0.11	1
Family adaptability	27.76	9.58	34.48	9.73	32.47	<0.001	0.10	1
Family satisfaction	3.34	1.30	4.02	1.15	21.32	<0.001	0.07	0.996
Personal well-being	48.39	12.42	57.49	8.80	54.77	<0.001	0.16	1

**Table 2 children-10-01057-t002:** Logistic regression results for emotion regulation difficulties, family functioning, and well-being predicting SI and SP in sample with BPD traits.

		Suicidal Ideation						Suicidal Planning				
	B	Wald X^2^	*p*	OR	95% CI (Low, High)	1-β	B	Wald X^2^	*p*	OR	95% CI (Low, High)	1-β
Gender (male)		0.04	0.98					2.99	0.22			
Female	−18.35	0.00	0.99	0.00	(0.00, 0.00)	1	−20.86	0.00	0.99	0.00	(0.00, 0.00)	1
No answer gender	−18.49	0.00	0.99	0.00	(0.00, 0.00)	1	−19.65	0.00	0.99	0.00	(0.00, 0.00)	1
Age	0.11	0.65	0.42	1.11	(0.86, 1.44)	0.07	0.06	0.20	0.65	1.06	(0.83, 1.34)	0.06
Awareness	−0.007	0.01	0.94	0.99	(0.82, 1.20)	0.05	−0.02	0.05	0.82	0.98	(0.84, 1.15)	0.05
Impulse	0.07	0.61	0.43	1.07	(0.90, 1.28)	0.06	−0.09	1.49	0.22	0.91	(0.79, 1.06)	0.06
Non-acceptance	0.12	3.48	0.06	1.13	(0.99, 1.28)	0.08	0.02	0.23	0.63	1.02	(0.93, 1.13)	0.05
Goals	0.11	1.37	0.24	1.12	(0.93, 1.36)	0.07	0.03	0.16	0.69	1.04	(0.88, 1.22)	0.05
Clarity	0.20	3.78	0.05	1.23	(1, 1.50)	0.13	−0.06	0.60	0.44	0.94	(0.80, 1.10)	0.06
Strategies	−0.05	0.28	0.60	0.95	(0.78, 1.15)	0.05	0.24	6.22	0.01	1.27	(1.05, 1.52)	0.14
Family cohesion	0.10	1.15	0.29	1.11	(0.92, 1.34)	0.07	0.03	0.16	0.69	1.03	(0.88, 1.21)	0.05
Family adaptability	0.08	0.58	0.45	1.08	(0.89, 1.31)	0.06	−0.04	0.29	0.59	0.96	(0.82, 1.12)	0.05
Family satisfaction	−1.34	6.06	0.01	0.26	(0.09, 0.76)	0.99	−0.39	0.84	0.36	0.68	(0.290, 1.56)	0.29
Personal well-being	−0.08	2.35	0.13	0.93	(0.84, 1.02)	0.06	−0.12	7.05	0.008	0.89	(0.81, 0.97)	0.07

**Table 3 children-10-01057-t003:** Logistic regression results for emotion regulation difficulties, family functioning, and well-being predicting SA and NSSI in sample with BDP traits.

			Suicidal Attempts					Non-Suicidal Self-Injury				
	B	Wald X^2^	*p*	OR	95% CI (Low, High)	1-β	B	Wald X^2^	*p*	OR	95% CI (Low, High)	1-β
Gender (male)		0.31	0.86					2.27	0.32			
Female	0.09	0.00	0.95	1.09	(0.06, 19.91)	0.06	1.55	1.34	0.25	4.69	(0.34, 64.01)	0.99
No answer gender	−0.27	0.04	0.85	0.76	(0.05, 11.57)	0.20	1.86	2.10	0.15	6.39	(0.52, 78.44)	0.99
Age	−0.04	0.15	0.70	0.96	(0.77, 1.20)	0.05	−0.02	0.04	0.85	0.98	(0.82, 1.18)	0.05
Awareness	−0.00	0.00	0.96	1.00	(0.87, 1.14)	0.05	0.01	0.03	0.86	1.01	(0.90, 1.14)	0.05
Impulse	0.17	4.63	0.03	1.19	(1.02, 1.39)	0.11	0.08	1.99	0.16	1.08	(0.97, 1.21)	0.06
Non-acceptance	0.03	0.39	0.53	1.03	(0.94, 1.13)	0.05	−0.07	2.93	0.09	0.93	(0.86, 1.01)	0.06
Goals	−0.10	1.17	0.28	0.91	(0.77, 1.08)	0.07	−0.04	0.31	0.56	0.96	(0.84, 1.10)	0.05
Clarity	0.06	0.65	0.42	1.07	(0.91, 1.24)	0.06	0.07	1.31	0.26	1.08	(0.95, 1.22)	0.06
Strategies	0.03	0.09	0.77	1.03	(0.86, 1.23)	0.05	0.08	1.32	0.25	1.08	(0.95, 1.24)	0.06
Family cohesion	0.03	0.17	0.68	1.03	(0.89, 1.19)	0.05	−0.02	0.12	0.73	0.98	(0.86, 1.11)	0.05
Family adaptability	−0.03	0.15	0.70	0.97	(0.83, 1.13)	0.05	−0.00	0.00	0.95	1.00	(0.88, 1.13)	0.05
Family satisfaction	−0.68	2.56	0.11	0.51	(0.22, 1.16)	0.73	−0.65	3.66	0.05	0.52	(0.27, 1.02)	0.73
Personal well-being	−0.03	1.23	0.27	0.97	(0.92, 1.02)	0.05	0.01	0.29	0.62	1.01	(0.97, 1.06)	0.05

## Data Availability

All materials in this article are available upon request from the authors.

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
