# Peer review of "Emotion Regulation Difficulties, Family Functioning, and Well-Being Involved in Non-Suicidal Self-Injury and Suicidal Risk in Adolescents and Young People with Borderline Personality Traits"

_children, 2023, doi:10.3390/children10061057_

Round 1

Reviewer 1 Report

The present study investigates the prevalence and correlates of Borderline Personality Disorder (BPD) traits in sample of 285 young participants recruited in Tenerife.

The investigated topic is scientific and clinical interest. The study was conducted with sound methodology and the statistical analyses are appropriate for the investigated hypotheses. The manuscript, overall, is well written.

The manuscript could therefore represent a valuable contribution to the research field.

However, some revisions could improve the overall quality of the work and its interest for the reader.

Introduction:

Study aims and the main hypotheses of the research could be presented in a separate subparagraph of the Introduction section for better clarity.

Methods:

The “Participants” subsection of the Methods section should focus on describing the context, duration and methodology for the recruitment of participants. The qualitative and quantitative description of the investigated sample should be moved to the beginning of the Results section.

Dates of beginning and conclusion of the recruitment process should be explicitly reported to improve the reproducibility of the findings.

The program used to conduct the statistical analyses and its utilized version should be explicitly reported in the manuscript. The threshold of statistical significance should also be explicitly reported: both p<0.10 and p<0.05 are mentioned, and a non-standard choice of p<0.10 has to be explicitly motivated.

Results:

Introducing a dedicated column for p-values in Table 1, 2 and 3 rather than reporting this information only through the presence of asterisks would considerably improve the completeness of the results.

Discussion:

A relevant limitation of the present study is that ADHD symptoms were not explored in the sample. ADHD symptoms can sometime overlap with BPD traits, particularly in younger individuals, but have a very different treatment profile. Moreover, ADHD symptoms and BPD traits are often present as comorbid conditions (see Kooij JJS et al., Updated European Consensus Statement on diagnosis and treatment of adult ADHD. Eur Psychiatry. 2019. 56:14-34. doi: 10.1016/j.eurpsy.2018.11.001 and Valsecchi P et al., Adult ADHD: Prevalence and Clinical Correlates in a Sample of Italian Psychiatric Outpatients. J Atten Disord. 2021. 25(4):530-539. doi: 10.1177/1087054718819824). This issue should be mentioned and explicitly discussed in the manuscript.

A paragraph briefly detailing potential treatment options for significant BPD traits in younger individuals should be provided to the readers, describing both pharmacological and non-pharmacological approaches that could provide clinical benefits for specific treatment outcomes.

Author Response

Introduction:

Study aims and the main hypotheses of the research could be presented in a separate subparagraph of the Introduction section for better clarity.

Thank you for the suggestion. The objectives and hypotheses have been presented in a separate paragraph.

Methods:

The “Participants” subsection of the Methods section should focus on describing the context, duration and methodology for the recruitment of participants.

The reviewer is right that the context, duration, and methodology for recruiting participants has not been sufficiently specified. This information has been included in the procedure section.

The qualitative and quantitative description of the investigated sample should be moved to the beginning of the Results section.

We thought it appropriate to keep the description of the sample in the participants’ section so that the remaining sections on procedure or data analysis can be better understood.

Dates of beginning and conclusion of the recruitment process should be explicitly reported to improve the reproducibility of the findings.

The reviewer is right. Information on the recruitment process and data collection dates have been included in the procedure section.

The program used to conduct the statistical analyses and its utilized version should be explicitly reported in the manuscript.

The statistical program used has been reported in the data analysis section.

The threshold of statistical significance should also be explicitly reported: both p<0.10 and p<0.05 are mentioned, and a non-standard choice of p<0.10 has to be explicitly motivated.

Statistical significance is certainly considered to be below 0.05, so we have eliminated marginally significant results (p < 0.10). However, in the discussion we have maintained the debate about how the strategy of emotional dysregulation centered on the non-acceptance of negative emotional responses played a differential role in the prediction of suicidal ideation and NSSI.

Results:

Introducing a dedicated column for p-values in Table 1, 2 and 3 rather than reporting this information only through the presence of asterisks would considerably improve the completeness of the results.

We have accepted the reviewer’s comment and have included the p-values in the tables.

Discussion:

A relevant limitation of the present study is that ADHD symptoms were not explored in the sample. ADHD symptoms can sometime overlap with BPD traits, particularly in younger individuals, but have a very different treatment profile. Moreover, ADHD symptoms and BPD traits are often present as comorbid conditions (see Kooij JJS et al., Updated European Consensus Statement on diagnosis and treatment of adult ADHD. Eur Psychiatry. 2019. 56:14-34. doi: 10.1016/j.eurpsy.2018.11.001 and Valsecchi P et al., Adult ADHD: Prevalence and Clinical Correlates in a Sample of Italian Psychiatric Outpatients. J Atten Disord. 2021. 25(4):530-539. doi: 10.1177/1087054718819824). This issue should be mentioned and explicitly discussed in the manuscript.

We welcome the reviewer’s suggestion. BPD shows comorbidity with other personality disorders and ADHD symptoms. We have therefore included this issue in the study limitations.

A paragraph briefly detailing potential treatment options for significant BPD traits in younger individuals should be provided to the readers, describing both pharmacological and non-pharmacological approaches that could provide clinical benefits for specific treatment outcomes.

Thank you for the suggestion. This information has been added in the penultimate paragraph of the discussion.

Reviewer 2 Report

Dear Editor,
I really appreciate the opportunity to review the manuscript  entitled:
"Emotional Regulation Difficulties, Family Functioning and Well-Being involved in Non-Suicidal Self-Injury and Suicidal Risk in Adolescents and Young People with Borderline Personality Traits"

I commend the authors for describing this critical and timely issue. The paper is interesting and well-written; however, I would like to highlight some issues that merit revision:

The study was very well conducted and the results are clearly and accurately stated. My only comment concerns the lack of clarity because of the scales used concerning past diagnoses especially any current therapies or use of substances of abuse or alcohol. In fact, the interaction on the aspects taken into examination of both drug therapies and psychotropic substances is well known, and it would be good to indicate whether such data were examined more strongly. I beg the authors to emphasize this more strongly in the manuscript, or if the data is not available to include it in the limitations. Overall an excellent paper

Author Response

Thank you very much for your positive feedback.

Certainly, we do not record the use of substances or abuse of alcohol. In limitations, we have highlighted the comorbidity that can exist between borderline disorder and substance abuse.

Reviewer 3 Report

Thank you for the opportunity to review this paper. I would like to make a few suggestions.

Introduction - is there a rationale for choosing these particular independent variables? Since there are many factors predicting suicidal behaviors, why these particular variables? Would the authors like to propose the theoretical framework that guided their choice? ll. 58-61 briefly mentioned the  motivational–volitional model. Is this suitable to be the guiding theory?

For each study that has been cited to provide literature review, please provide the context in which those studies were conducted.

Since the study involved both adolescents and young adults, the authors could provide prevalence of BPD and BPD traits, SA, SI and SP in individuals from the young adult age range as well.

Paragraph has also focussed on adolescents. How about young adults?

l. 69 " such high-risk behaviors" - May I know which high-risk behavior is being referred to here?

Methods

Is the Personal Well-being Index School Version appropriate for young adults who may be working or in college?

Is there a reference for the The family satisfaction scale? Is it developed by the authors, or was it derived from another study? Since it is only one item, perhaps the authors could provide the question.

ll. 143-144 - "The Spanish adolescent version obtained the following Cronbach’s alpha". How about the adult version?

Procedures - Several state schools were invited. How were they selected? How may were invited, and how many declined? How about the young adults? Where were they recruited?

Sample size - how was sample size determined? Could the authors conduct a post-hoc analysis to justify 103 samples to conduct the logistic regression, is it of enough power? Or are the results non-significant because of not enough power?

l. 189 - adolescents and young adults?

Table 2 - What do Gender (1) and Gender (2) refer to? There are a few spelling errors, such as 95% CI. BPD (not BDP), and X square, the 2 should be superscripted.

The authors use p<.10. Usually health and social science studies use a significance level of 0.05. For Table 3, especially for the result of non-acceptance x NSSI, 95%CI was (.86, 1.01). What is the rationale for using 0.1? I would like to recommend the cutoff of p<0.05.

In the first paragraph of the discussion, the authors summarised the study. However, it was too overgeneralised. Please provide a more detailed summary of the study, ie., which aspects of emotional regulation difficulties contributed to which suicidal behavior or NSSI. Some of the associations are just marginal, and therefore to claim "key mechanism" for them seem like an overstatement.

Discussion and conclusions: May I recommend that the authors discuss and conclude only the results of their study, and not overgeneralise. For example, the study did not investigate interpersonal relationships, but family adaptability, cohesion, and satisfaction, and only family satisfaction was found to predict NSSI. This principle should be applied throughout.  Thank you.

Minor editing needed.

Author Response

Thank you very much for your suggestions.

Introduction - is there a rationale for choosing these particular independent variables? Since there are many factors predicting suicidal behaviors, why these particular variables? Would the authors like to propose the theoretical framework that guided their choice? ll. 58-61 briefly mentioned the motivational–volitional model. Is this suitable to be the guiding theory?

The reviewer is right that we have not adequately clarified the choice of emotional and familial variables. At the beginning of the third paragraph, we have added Lineham’s model, which guided us in this choice, given that findings indicate that emotional regulation and family relationships are mainly affected in people with BPD. However, the suicide and NSSI models share with the explanatory model of BPD the emotional and interpersonal components that trigger these self-harming behaviors.

For each study that has been cited to provide literature review, please provide the context in which those studies were conducted.

We have reviewed the literature and clarified the population characteristics involved in the various studies cited. We hope we have understood the reviewer’s suggestion.

Since the study involved both adolescents and young adults, the authors could provide prevalence of BPD and BPD traits, SA, SI and SP in individuals from the young adult age range as well. Paragraph has also focussed on adolescents. How about young adults?

We thank the reviewer for this contribution. Information on the prevalence of BPD, suicide, and NSSI in young adults has been provided in the first paragraph of the introduction.

  1. 69 “ such high-risk behaviors” – May I know which high-risk behavior is being referred to here?

We have used the term “high-risk behaviors” to refer to suicide and NSSI, but given that there are other risk behaviors (substance abuse, alcohol, risky sexual behavior, reckless driving…), we have decided to remove the term throughout the manuscript to avoid confusion.

Methods

Is the Personal Well-being Index School Version appropriate for young adults who may be working or in college?

When comparing both child/adolescent and adult versions, no substantial differences appear since the same vital areas (standard of living, personal health, achievement in life, personal relationships, personal safety, feeling part of the community, future security) are assessed. However, the items have been better clarified; for example, the adult version contains the question “How satisfied are you with your personal relationships?” and in the child version the question is “How happy are you about getting on with the people you know?”

The equivalence between the two scales has been tested by the following authors:

Tomyn, A. J., Fuller Tyszkiewicz, M. D., & Cummins, R. A. (2013). The personal wellbeing index: psychometric equivalence for adults and school children. Social Indicators Research, 110, 913-924.

Is there a reference for the The family satisfaction scale? Is it developed by the authors, or was it derived from another study? Since it is only one item, perhaps the authors could provide the question.

Thank you for the suggestion. We have included the item used to measure satisfaction with the family.

  1. 143-144 - "The Spanish adolescent version obtained the following Cronbach’s alpha". How about the adult version?

Both the adult and adolescent versions have adequate psychometric criteria. The required information has been included in the instrument paragraph.

Procedures - Several state schools were invited. How were they selected? How may were invited, and how many declined? How about the young adults? Where were they recruited?

The reviewer is right. This information has been added to the procedure section.

Sample size - how was sample size determined? Could the authors conduct a post-hoc analysis to justify 103 samples to conduct the logistic regression, is it of enough power? Or are the results non-significant because of not enough power?

Thank you for the recommendation. The post-hoc statistical power of the logistic regression analyses has been calculated using g power and included in the results section and in the tables, along with the limitation of the scope of the results due to sample size.

  1. 189 – adolescents and young adults?

The reviewer is right: young people have been included in the sentence where the chi-square results are described.

Table 2 - What do Gender (1) and Gender (2) refer to? There are a few spelling errors, such as 95% CI. BPD (not BDP), and X square, the 2 should be superscripted.

The gender variable is a nominal variable that consisted of three categories: 0 male, 1 female, 2 prefer not to answer. We therefore created two dummy variables (k-1) in which the reference value was male, value 1 corresponded to female, and value 2 to those who prefer not to answer. In the tables, gender 1 and gender 2 have been replaced by the corresponding gender to avoid confusion. Thank you for detecting the errors relating to 95% CI and chi-square, they have been rectified.

The authors use p<.10. Usually health and social science studies use a significance level of 0.05. For Table 3, especially for the result of non-acceptance x NSSI, 95%CI was (.86, 1.01). What is the rationale for using 0.1? I would like to recommend the cutoff of p<0.05.

Indeed, the significance level in health and social science is below 0.05, so we have eliminated marginally significant results (p < 0.10). However, in the discussion we have kept the debate about how the strategy of emotional dysregulation centered on the non-acceptance of negative emotional responses played a differential role in the prediction of suicidal ideation and NSSI. For suicidal ideation the p was 0.06 and for NSSI, .09.

In the first paragraph of the discussion, the authors summarised the study. However, it was too overgeneralised. Please provide a more detailed summary of the study, ie., which aspects of emotional regulation difficulties contributed to which suicidal behavior or NSSI. Some of the associations are just marginal, and therefore to claim "key mechanism" for them seem like an overstatement.

The reviewer is right that the first paragraph is very general, but the intention was to highlight the differences between adolescents and young people with BPD traits and adolescents without this vulnerability. It has been clarified in the paragraph. We have also replaced “key mechanism” with “relevant factors”. Furthermore, following the reviewer’s suggestions, we have clarified the concrete variables predicting the different types of suicidal and NSSI behavior.

Discussion and conclusions: May I recommend that the authors discuss and conclude only the results of their study, and not overgeneralise. For example, the study did not investigate interpersonal relationships, but family adaptability, cohesion, and satisfaction, and only family satisfaction was found to predict NSSI. This principle should be applied throughout.

Thank you for the suggestion. We sometimes used the terms interpersonal or social relationships interchangeably with family relationships or family functioning. The manuscript has been revised to clarify this issue, taking into account that family relationships also involve interpersonal relationships, but that they are not exclusive.

Round 2

Reviewer 1 Report

The Authors have responded appropriately to all queries.

Reviewer 3 Report

Thanks for the second opportunity to review this manuscript. The authors have addressed all concerns raised. Perhaps minor editing of English language is required. Thank you very much.

Minor editing for some errors are still needed.